# Robotic Surgery: Is There a Possibility of Increasing Its Application in Pediatric Settings? A Single-Center Experience

**DOI:** 10.3390/children9071021

**Published:** 2022-07-08

**Authors:** Edoardo Bindi, Camilla Todesco, Fabiano Nino, Giovanni Torino, Gianluca Gentilucci, Giovanni Cobellis

**Affiliations:** 1Pediatric Surgery Unit, Salesi Children’s Hospital, 60123 Ancona, Italy; todesco.camilla94@gmail.com (C.T.); fabiano.nino@ospedaliriuniti.marche.it (F.N.); giovannitorino1@libero.it (G.T.); giallygent@gmail.com (G.G.); giovanni.cobellis@ospedaliriuniti.marche.it (G.C.); 2Dipartimento di Scienze Cliniche Specialistiche ed Odontostomatologiche, Università Politecnica of Marche, 60020 Ancona, Italy

**Keywords:** robot-assisted surgery, minimally invasive surgery, robotic learning curve

## Abstract

**Introduction:** Robotic surgery has shown explicit benefits and advantages in adults, but it is not yet strongly established in the pediatric population, even though its popularity is increasing, especially in the urologic field. **Materials and methods:** In this article we present our experience with the Da Vinci System (SI first and XI nowadays) at our pediatric institution in order to analyze our progress over the years. We considered all patients from the start of the robotic surgery program in 2016 until the end of 2021, dividing them into general abdominal surgery and genitourinary surgery. Analyzed data were the patient’s demographic, details of surgery, and intra and post-operative complications. **Results:** The total number of patients (pts) included in this study was 76, of whom 40 (52%) were male and 36 (48%) were female. The mean age at surgery was 90.9 months (range 10–207 months), and the mean weight at surgery was 29.3 kg (range 9.5–68 kg). There were 18 general abdominal robotic surgeries and 58 genitourinary robotic surgeries performed. The most performed surgeries in these two categories were fundoplication for gastro-oesophageal reflux disease (11%) and Anderson–Hynes pyeloureteroplasty. The mean operative time was 224.2 min (range 72–530 min): the mean times in the two groups (general abdominal surgery and genitourinary surgery) were 165 min (range 84–204 min) and 194 min (range 95–360 min), respectively. A total of four (5%) minor complications were reported. The total conversions were two (2.6%) and the mortality rate was 0%. **Conclusions:** Pediatric robotic surgery is a field of considerable interest and it is rapidly expanding. In our experience, it is evident how the learning curve has increased gradually, but steadily, allowing us to advance from standardized surgery, such as fundoplication and pieloplasty, towards a more technically complex one, achieving the same good results. We believe that robotic surgery is very respectful of tissues and feasible, especially for reconstructive surgery. For these reasons, it could become of common use also in the pediatric population, overcoming impediments such as excessive cost and the lack of pediatric instruments, in order to be able to treat children with a progressively lower age and weight.

## 1. Introduction

Charles Darwin, in his 1859 masterpiece titled “The Origin of Species”, stated that it is not the strongest of the species that survives, but the most responsive to change. In the same way, this law could also be applied to surgery, which has evolved greatly over recent centuries to the improvements achieved in the present day.

One of the most important steps in this growth process has been the birth and growth of minimally invasive surgery (MIS), which has massively changed the surgical approach, first in adults, and later in the pediatric population [1]. In fact, with small instruments and better magnification with a camera, it has been possible to perform a large amount of surgery with better results, such as less postoperative pain, a shorter hospitalization time, and better cosmetic outcome.

Today, the introduction of robot-assisted surgery is another passage in the evolutionary process of surgery. This technology offers significantly improved three-dimensional (3D) visualization and instrumentation dexterity combined with tremor cancellation, enabling surgeons to perform complex interventions of reconstructive surgery [2]. Robot-assisted surgery with these advantages has surpassed the laparoscopic approach in many surgical fields, becoming, for some surgical branches such as urology and general surgery, a gold standard for many procedures [3]. In pediatric surgery, however, this type of approach has not had the same wide adoption because of the difficulty of finding the right indications in such a young population. Indeed, the use of the robot in children has found limitations due to the need to work in a limited space and small cavities with instruments developed for the adult, but also some specific difficulties such as patient and trocar position, anesthesia, and postoperative pain control [4,5].

However, in the early 2000s [6], robotic surgery began to be applied in pediatric procedures as well, and slowly, in the last decade, it has begun to be a first choice in selected procedures [7].

In this paper, our aim was to highlight the state of the art of pediatric robotic surgery in our center, evaluating its growth and development, and comparing our experience with the most important case reports in the literature.

## 2. Materials and Methods

The study was conducted at the Pediatric Surgery Department of Salesi Childrens Hospital, in Ancona. A retrospective study was performed on children undergoing surgery with robotic technique from November 2016 to November 2021. The Da Vinci robotic system was used in all cases, Si and Xi before and after July 2019, respectively. We divided the surgeries into two groups based on the anatomical district of interest: general abdominal robotic surgery and genitourinary robotic surgery. Data were collected on patient characteristics, diseases treated, surgical procedures performed, and complications. The complications were divided into minors and majors on the basis of Clavien–Dindo Classification. We considered a complication as minor if it was grade I/II, and as major if grade III/IV/V on the basis of Clavien–Dindo Classification.

### Surgery

All surgical procedures were always performed according to the same principles and following the pattern we have adopted since the beginning of our experience with the Da Vinci system.

We always used three robotic trocars: one 12-mm trocar for optics (inserted at the umbilical level with an open technique) and two 8-mm trocars. In addition, a 5-mm laparoscopic trocar, using the surgeon at the operating table as an aid, was used. The robotic trocars were placed on the same line with a distance of about 4 fingers from each other.

The placement of the trocars was always done with standard patterns for each type of surgical procedure, adopting minimal variations in selected cases.

Taking, as an example, the surgery that was performed in the largest number, namely pyeloureteroplasty according to Anderson–Hynes, the surgical preparation of the patient was done according to the following description. The patient was placed on the operating table in left lateral decubitus at a 60-degree angle from the operating table. The patient was placed on the edge of the operating table, effectively on the edge of the operating table opposite the da Vinci Robot trolley. One 12-mm robotic trocar was used in the umbilicus (open technique access by transumbilical route) and two 8-mm robotic trocars were used in the right iliac fossa and epigastrium, respectively (the trocar in the right iliac fossa was placed 1 cm superior to the spino-umbilical line at 3.5 fingers from the umbilicus measure taken at an inflated abdomen, i.e., with pneumoperitoneum at 10/11 mmHg, while the trocar in epigastrium was placed 1 cm to the right of the xifo-umbilical line at 4 fingers from the umbilicus measure taken at an inflated abdomen, i.e., with pneumoperitoneum at 10/11 mmHg). In addition, a 5/6 mm accessory laparoscopic trocar placed on the midline (xifo-umbilical) midway between the umbilical trocar and the robotic epigastric trocar measurement taken at inflated abdomen, i.e., with pneumoperitoneum at 10/11 mmHg was used. Finally, a 30-degree 12-mm robotic optic was used.

In our opinion, for the size of our patients, this scheme was effective, and we never experienced difficulties due to collisions between instruments.

## 3. Results

The total number of patients (pts) included in this study was 76, of whom 40 (52%) were male and 36 (48%) were female. The mean age at surgery was 90.9 months (range of 10–207 months), and the mean weight at surgery was 29.3 kg (range of 9.5–68 kg). Patient demographics are shown in Table 1.

There were 18 (24%) general abdominal robotic surgeries and 58 (76%) genitourinary robotic surgeries performed (Figure 1).

The most performed surgeries in these two categories were fundoplication for gastro-oesophageal reflux disease (11%) and Anderson–Hynes pyeloureteroplasty for hydronephrosis due to the obstruction of the pyeloureteral junction (46%), respectively.

The procedures performed in general abdominal surgery were as follows (Figure 2):−Nissen fundoplication for gastroesophageal reflux disease: nine pts (50%)−Hiatoplasty for recurrent hiatal hernia: one pt (6%)−Removal of ovarian neoformation: two pts (11%)−Removal of gastric cystic duplication: two pts (11%)−Excision of hepatic cyst: one pt (6%)−Left adrenalectomy for pheochromocytoma: one pt (6%)−Excision of choledochal cyst and hepatic jejunanastomosis: two pts (11%) (Figure 3).The procedures performed in genitourinary surgery were as follows (Figure 4):−Anderson–Hynes pyeloureteroplasty for hydronephrosis due to obstruction of the pyeloureteral junction: thirty-five pts (60%); in two of these, lithotripsy was associated due to the presence of renal stones (Figure 5)−Ureterocalicostomy for hydronephrosis due to recurrent pyeloureteral junction obstruction: one pt (2%)−Vascular Hitch (Hellstrom suspension of abnormal renal polar vessels) for hydronephrosis due to obstruction of the pyeloureteral junction: one pt (2%)−Lich–Gregoir ureteral reimplantation for vesicoureteral reflux: 14 pts; exeresis of a bladder diverticulum was performed in two of them (24%) (Figure 6)−Upper double renal district heminephrectomy: three pts (5%); in one case, it was bilateral for urinary pseudoincontinence from ectopic upper ureters.−Upper partial nephrectomy for epithelioid cell tumor: one pt (2%)−Mitrofanoff appendico-vesicostomy in previous bladder exstrophy: one pt (2%)−Removal of ureteral polyp and ureteroanastomosis: one pt (2%)−Excision of atrophic vagina in male patient with disorder of sexual development: one pt (2%).

There has been a steady increase in the number of patients treated by robotic techniques in our center. Figure 7 shows the growth of robotic cases during our study period.

The mean operative time was 224.2 min (range of 72–530 min): the mean times in the two groups (general abdominal surgery and genitourinary surgery) were 165 min (range of 84–204 min) and 194 min (range of 95–360 min), respectively.

Considering the two most performed surgeries in the two groups, we see that the average operative time for fundoplication was 174 min (range 125–210 min) and 182 min (range of 170–250 min) for pyeloplasty. Looking at the operative times of these two surgeries, an interesting finding emerges, namely that, over the years, as the number of cases performed increased, the operative time of the single surgery gradually decreased (Figure 8 and Figure 9), thus demonstrating an improvement in the surgical learning curve.

A total of four (5%) minor complications were reported: one (1%) for abdominal surgery (one wound infection) and three (4%) for genitourinary surgery (one wound infection and two partial surgical wound dehiscences). There were two (2.6%) conversions, one in a case of genitourinary surgery (one heminephrectomy) and the other in a case of general abdominal surgery (one fundoplication in adhesion syndrome from previous surgery for necrotizing enterocolitis).

At a mean follow-up of 2.5 years, we recorded two cases of vesicoureteral reflux recurrence (2.6%); both patients underwent a reoperation of the transvesical ureteral reimplantation, according to Cohen, and the endoscopic infiltration of the refluent ureteral meatus, respectively.

No patient died during surgery, in the post-operative course, or during follow up. Thus the recorded mortality rate was 0%.

## 4. Discussion

Robotic surgery began as an aid in military settings to rescue injured soldiers, and only later, in the early 1990s, was it introduced into the medical field [8]. AESOP^®^ (Automatic Endoscopic System for Optimal Position; Computer Motion, Inc., Goleta, CA, USA) was the first system to be approved by the Food and Drug Administration (FDA). It consists of a voice-controlled robotic arm that actively manipulates the telescope/camera, eliminating the need for human support for the camera and the associated difficulties in directing camera positioning [9]. In 2000, the FDA approved the use of the Da Vinci system and it has since been used in many institutions in thousands of surgical procedures. This technology is a teleoperative system consisting of a console for the surgeon and a side cart for the patient. The console is designed to accommodate the first operator who, through manual controllers, directs the movements of the robotic arms during surgery. From here, in addition, the surgeon controls the camera movements and can also control the interface panel. This system allows the remote control of the patient-side tower structure that consists of two or three arms that control the operating instruments, and a separate arm that controls the video endoscope [10].

This last class of robots has been used in the fields of general surgery, urology, gynecology, and cardiothoracic surgery, but only later and much more slowly in pediatric surgery. In fact, the size and variety of available robotic instrumentation remains limited compared to that offered for adult surgery, and the huge size discrepancy between pediatric patients and the overall robotic system’s size may limit surgical indications [11]. In 2002, Heller et al. described the first time of use of robotic surgical systems for abdominal procedures in children [12]. They reported the robotic approach, using a Da Vinci system, for performing a surgical intervention for the treatment of gastroesophageal reflux disease in 11 pediatric patients who underwent Thal or Nissen fundoplication. The mean age was 12 years old and they reported no complications in this work. Since then, the use of robotic surgery in the pediatric field has made important steps, and the indications for the procedure have been extended to other diseases and to patients of a lower age and weight. A retrospective study in 2019 [13] showed that weight cannot be considered an absolute limitation for robotic surgery. This finding was also shown to be true in our study, in which the smallest patient undergoing surgery was 10 months old with a weight of 9.5 kg. Improved instruments, in fact, allow complex surgical procedures to be performed in low-weight children without additional difficulty. Other studies [14,15] have also reported case series demonstrating the safety and feasibility of the robot in pediatric surgery.

The results of our study confirmed this growth process: a progressive increase in the number and complexity of surgeries each year is evident in our center. This result not only showed an increase in pediatric surgical indications, but also an improvement in the learning curve.

The learning curve of a new technique is fundamental to its application and represents the improvement in its performance over time. To date, the learning curve for the robot-assisted technique in surgery is evaluated on the basis of the reduction in operative time; in particular, the parameters taken into consideration are the time to prepare and activate the system, position, and attach the robot, and the time required to complete the procedure. Several studies have been done from which it can be seen that the surgeon’s experience allows, in a short time, to shorten the surgical time, a factor indicative of learning. Among them: in the study by Knight et al. [16], the authors performed 15 fundoplications with robotic technique and evaluated the operating time, which dropped from 323 min in the first case to 180 for the last; moreover, of these, 10 were performed by the same surgeon who showed a learning curve with a reduction in the time taken to complete the procedure from 251 to 64 min in the 10th case. In the study by Meehan et al. [17], when comparing 50 cases of robotic fundoplication, in the first five performed by the same surgeon, the total operative time decreased from 3 h to 90 min. Similarly, in the 2017 review by Binet et al. from the comparison of 60 fundoplications performed with the robotic techniques from two university pediatric surgery centers, there was a flattening of the learning after the 20th case, with a significantly decreased mean operative time.

This finding, which we believe is very important, is also confirmed by our study. In fact, in the 6 years of robotic experience, the operative timing gradually decreased from 245 min to 125 min, finding full correspondence with the elements that emerged from the studies mentioned above. In the same way we can talk about pyeloplasty which, according to our results, demonstrated a decrease in the operative time (from 250 to 170 min) during the same period.

These studies also show that the learning curve of this technique is very short; few surgeries are needed to achieve significant results. Reducing the operative time has no implications in terms of reducing complications or improving patient outcome in fundoplication surgery, but being, to date, indicative of learning the robotic technique, it consequently, becomes an expression of the acquisition of skill and experience by the surgeon and improvement in surgical performance. This is important considering the possibility that, in the future, the additional dexterity and advantages that the robot provides will pave the way for the use of this technique for increasingly complex procedures.

In fact, robot-assisted surgery remains an excellent aid in all those surgeries in which the reconstructive part is a key step in the execution of the surgical procedure. In our experience, we have observed the greatest advantages in reconstructive surgeries, such as pyeloureteroplasty, the removal of choledochal cysts with hepaticojejunanastomosis, ureteral reimplantation for vesicoureteral reflux, and Mitrofanoff appendico-vesicostomy for urinary incontinence. This concept is supported by much data in the literature. In 2019, Esposito et al. [18] compared the results of pyeloplasty performed laparoscopically and with robots, showing that robot-assisted pyeloplasty had higher success rates and that this technique was effective in treating recurrences previously operated with an open or laparoscopic technique. Per regards to other procedures, such as the treatment of a choledochal cyst, experience is more limited, both speaking of our own and of work in the literature. What does emerge is that a choledochocystectomy for children completely by robotic surgery and a Roux-en-Y hepaticojejunostomy is safe and feasible. In our experience, this approach has a clearer field than the open or laparoscopic technique, and the surgery is more accurate, and the injury is smaller. This results in less hospitalization time, less blood loss, and surgical wounds with a better cosmetic outcome.

Robotic surgery, compared with open surgery and laparoscopy, is also shown to be better for the surgeon. In fact, especially in more complex procedures, these latter two increase the physical fatigue of the operator, which, in longer surgeries, result more in an increased risk of making mistakes. The robotic technique has overcome these limitations, improving ergonomics and reducing the physical impact on operators.

These data are in line with the results shown by one of the most important studies on this topic. This 2013 [7] systematic review showed an increase through the years in both volumes of pediatric cases treated and the published literature on this topic. In addition, another interesting result of this review is the percentage of each procedure performed for the main anatomical district (abdomen in general and genitourinary system). The main surgical procedures performed with the robot are fundoplication for gastro-oesophageal reflux disease and pyeloureteroplasty for hydronephrosis due to the obstruction of the pyeloureteral junction. This is in agreement with what has been demonstrated in our work, showing a shared consensus in considering robotic surgery as a viable alternative for the treatment of gastroesophageal reflux disease and pyeloureteral junction obstruction. In fact, several studies [19,20] have shown no significant differences in the outcomes and complications between laparoscopy and the robotic technique in the approach to these pathologies in pediatric age. Additionally, it is evident in our case series that two of the most frequently performed procedures are gastric duplication excision and heminephrectomy. In past years, there was no evidence that the robot was the suggested approach for these types of surgeries, due to the fact that neither gastric duplication excision nor a heminephrectomy had a reconstructive part. In these cases, although it does not present a significant advantage, the use of robotic surgery has proven to be a novel and effective way in the minimally invasive performance of these surgical procedures [21,22].

In our experience, the application of the robot on a limited and very selected number of cases is explained by the fact that pediatric surgeons share the robotic system with general surgeons and urologists, which makes it possible to lower the costs associated with the use of this technology [23].

In our case report, we also reported data on ureteral reimplantation. Ureteral reimplantation is the surgical correction method for a pediatric vesicoureteral reflux (RVU). The laparoscopic approach for RVU was introduced as early as 1993, and the robotic approach was introduced about ten years later. Although open uretero-vesical reimplantation is still the gold standard in the treatment of pediatric RVU, the application of robot-assisted ureteral reimplantation has been increasingly adopted [24,25,26]. With the development of robotic instrumentation, this technique has been applied in clinical practice and has been shown to reduce postoperative pain, and shorten the recovery phase and hospitalization time. In addition, the robotic reimplantation technique has a shorter learning curve than the conventional laparoscopic approach [27].

Based on our experience, we consider the robotic approach for ureteral reimplantation to be safe and effective, with an incidence of recurrence overlapping with endoscopic and open techniques. Paying the price of a longer operative time, the robotic technique provides better visualization of anatomic structures, reducing the risk of damage to peri-ureteral and peri-vesical nerves.

A further field of the application of robotics is thoracic surgery, with no operations performed in our case series and a small number of cases reported in the literature. This finding can be explained by several reasons. First, in the pediatric population, many of the thoracic diseases are congenital (esophageal atresia, CPAM, lobar emphysema, pulmonary sequestration) and need surgery in the neonatal period or in the first months of life, making the use of the robot almost impossible so far. The other reason is that, in any case, even in older children, the thoracic cavity is small, making both the trocar placement and instrument movements difficult. Considering a possible further evolution of surgical robotic technology, the problem of interventions in thoracic and neonatal surgery is one of the limitations that could be overcome with the implementation of smaller and less impactful instruments on the anatomical structures of the pediatric patient.

## 5. Conclusions

The results of this work demonstrate that in our pediatric surgical center, as in the rest of Europe and the United States, robotic surgery is a field that has shown progressive growth. The presence of a significant minimally invasive surgical background facilitates learning the robotic technique. Pyeloureteroplasty and fundoplication are, to date, the most frequently performed basic surgeries in children, and those in which the results are at least equivalent to laparoscopic procedures, also being able to represent an important step in the training of younger people. As our experience has also shown, many other more complex procedures can be performed, even with greater advantages over basic techniques. A further increase in the learning curve and technological improvements, coupled with a desirable reduction in costs, may expand the application of robotic surgery in the pediatric age, and probably also in the neonatal age.

## Figures and Tables

**Figure 1 children-09-01021-f001:**
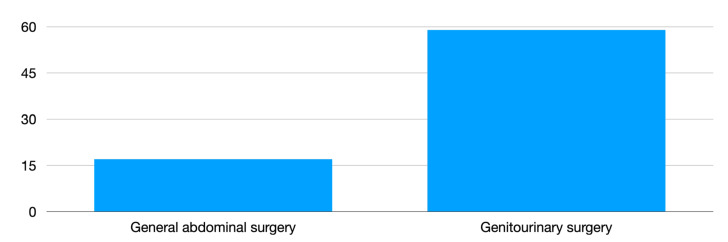
Number of surgeries by anatomical district.

**Figure 2 children-09-01021-f002:**
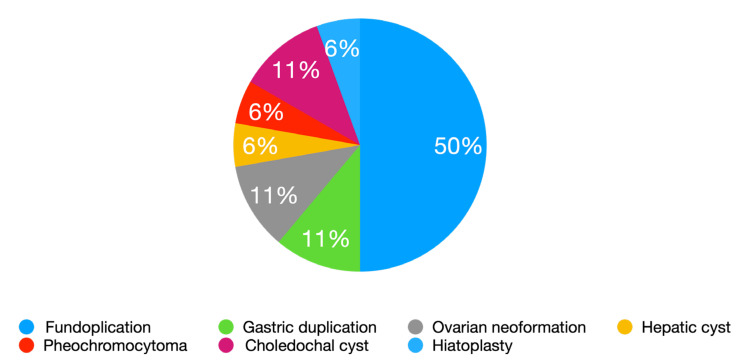
Major surgeries in General Abdominal Surgery.

**Figure 3 children-09-01021-f003:**
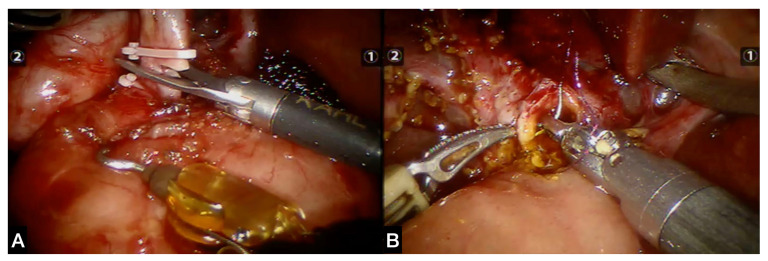
(**A**) Robotic exeresis of choledochal cyst and (**B**) subsequent hepaticojejunanastomosis.

**Figure 4 children-09-01021-f004:**
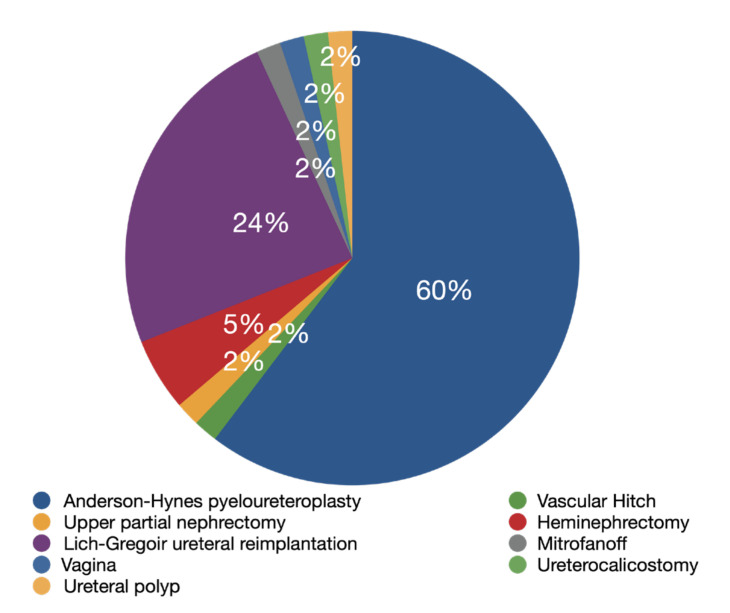
Major surgeries in Genitourinary Surgery.

**Figure 5 children-09-01021-f005:**
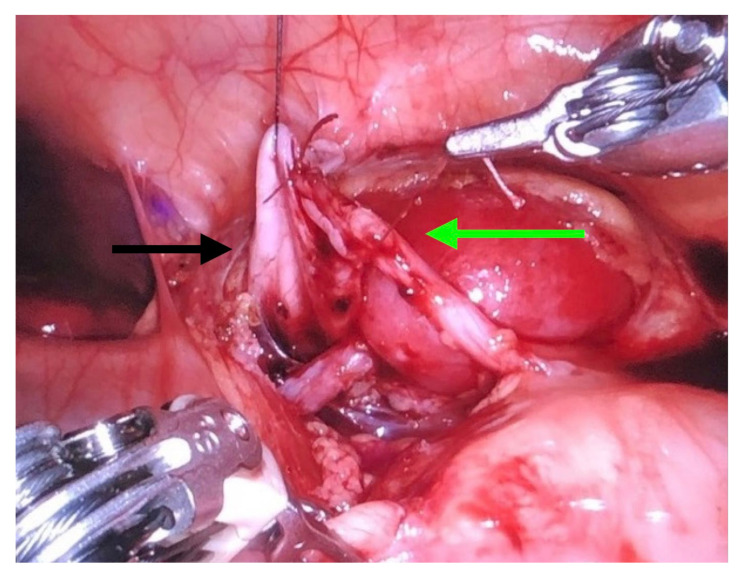
Robotic pyeloureteroplasty: renal pelvis (**black arrow**) and ureter (**green arrow**).

**Figure 6 children-09-01021-f006:**
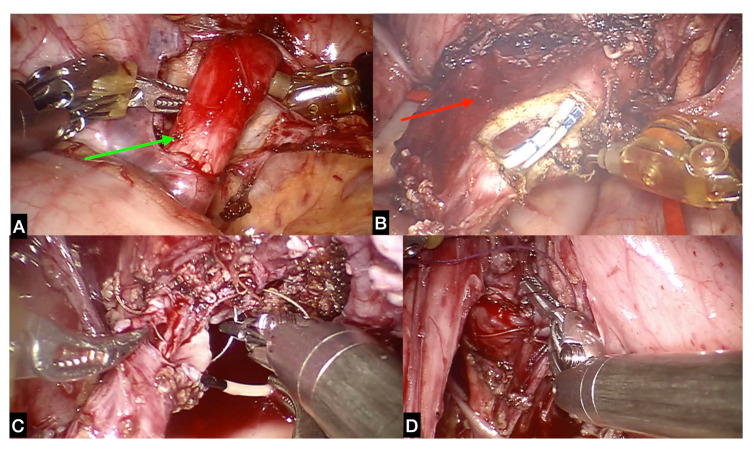
Ureteral reimplantation in patient with bladder diverticulum. (**A**) The ureter (green arrow) is isolated and (**B**) detached from the bladder with exeresis of the diverticulum (red arrow); (**C**) uretero-vesical anastomosis and (**D**) extravesical reimplantation according to Lich–Gregoir is then performed.

**Figure 7 children-09-01021-f007:**
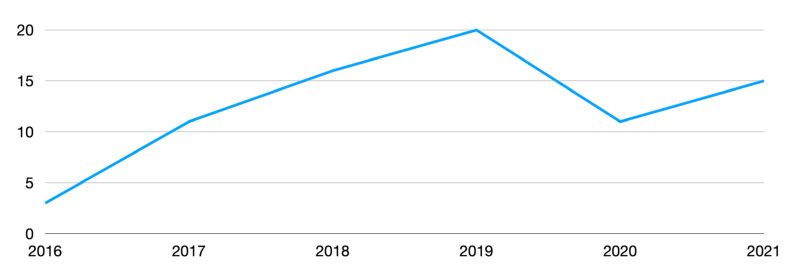
Increase in cases performed by robotic technique during the study period (the decrease in early 2020 is due to the SARS-CoV-2 emergency that led to the cancellation of elective surgeries for a period).

**Figure 8 children-09-01021-f008:**
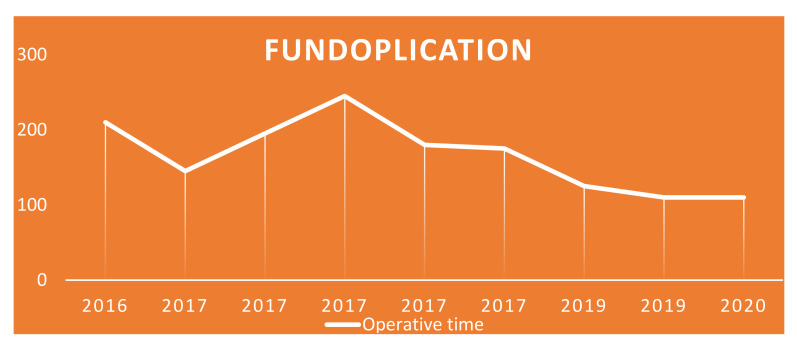
Fundoplication: trend of operating times over the years.

**Figure 9 children-09-01021-f009:**
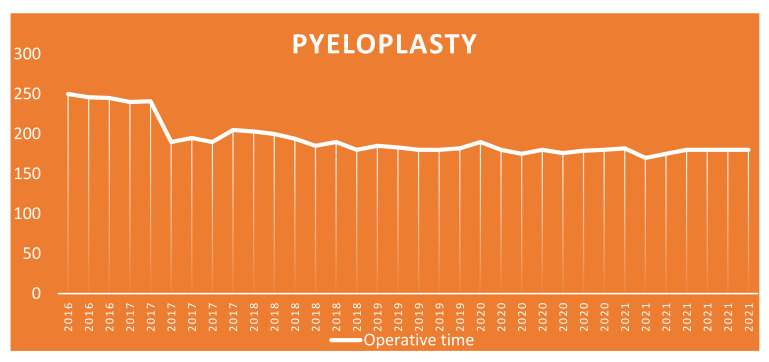
Pyeloplasty: trend of operating times over the years.

**Table 1 children-09-01021-t001:** Study population data.

Patients	76
Males/Females	40 (52%)/36 (48%)
Mean age at surgery	90.9 months (range 10–207 months)
Mean weight at surgery	29.3 kg (range 9.5–68 kg)
Mean operative time	224.2 min (range 72–530 min)
Mean hospital stay	3.7 days (range 2–12 days)
Major complications	0
Minor complications	4 (5%)
Conversions	2 (2.6%)

## Data Availability

No new data were created or analyzed in this study. Data sharing is not applicable to this article.

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
