# Peer review of "Robotic Surgery: Is There a Possibility of Increasing Its Application in Pediatric Settings? A Single-Center Experience"

_children, 2022, doi:10.3390/children9071021_

Round 1

Reviewer 1 Report

The authors describe a relatively large case series of robotic operations performed in a pediatric population. This is useful data to add to the literature on the safety and feasibility of robotic surgery in children. My critiques are as follows:

1. The Results section of the abstract is lacking. I would include the length of stay, operative time, conversion rate, and complications reported.

2. The opening paragraph in the introduction is unnecessary. There is extensive literature regarding the benefits of minimally invasive and then robotic surgery in adults and the next logical step is to apply robotic surgery to children.

3. The authors do describe the complications in the body of the Results section but major and minor complications should be defined in the Methods section. In addition are there any other technical aspects that can be included either in the Methods or Discussion - were the same ports used as with adults? How was positioning/docking performed to minimize arm collisions in a smaller space?

4. There are several English language or typographical/editorial errors, for example "fundoplication" is listed several times including in the figure as "fundoplicatio".

5. The figures with intraoperative images panels are useful but to improve I would add greater detail on what is being displayed in the figure legends. The use of arrows/arrowheads can be helpful in highlighting the main point being displayed. For images with multiple panels I suggest describing each separately as A, B, C, etc

6. What was the operative time for the most commonly performed procedures (fundoplication and pyeloureteroplasty)? It is less useful to compare the operative time across a very heterogenous group of operations, for example choledochal cyst excision with hepaticojejunostomy and removal of ovarian neoformation.

7. The authors show increased use over time and describe the learning curve of using the robot. Is there any evidence of improved learning curve, for example, decreased operative time with more experience?

8. The main question with this manuscript that needs to be addressed in the Discussion is why should the robotic be used and expanded in pediatric surgery. The authors mention and cite a reference showing no difference in laparoscopic vs robotic fundoplication, so a discussion on the benefits of the robot should be included for procedures that are already commonly performed laparoscopically. Are the ergonomics better for the surgeon? Is cost better or worse with robot? Is blood loss, complication rate, or recurrence rate lower, or any difference in conversion rate to open?

A strength of the manuscript is the use of the robotic technique for complex operations requiring reconstruction (for example, the choledochal cyst excision with hepaticojejunostomy). I would highlight this strength in greater detail in the Discussion, compare the length of stay and complication rate reported for these procedures to open approach to the same procedures and comment that the robot can expand access to minimally invasive techniques and therefore faster recovery time.

Author Response

Dear reviewer,

thanks for the useful suggestions.

We did the corrections on the base of your indications (each change is highlighted in red).

  1. We added some data in the Result's section of the abstract.
  2. Although the statement in the introduction are well-known we believe that these are still useful. So we didn't delete them.
  3. We added the requested data.
  4. We corrected the wrong words.
  5. We modified the figures.
  6. We added the requested data.
  7. We gave information about the learning curve, also adding two pictures.
  8. We added more informations in the discussion.

Reviewer 2 Report

This is a well written presentation of the robotic experience of a single institution in pediatric surgery. The paper is quite interesting. I have few observations:

1) the title mentions review of the literature but in the article there is a bare minimum in the discussion. I suggest to expand this part as it would be very interesting for the readers to gain deeper insight to the state of the art in pediatric robotic surgery, or alternatively remove review of the literature form the title. 

2) the davinci system was not FDA approved in 1995, please verify your information and correct the statement.

3) you should have some sort of IRB approval or an equivalent form your country for this type of study.

Author Response

Dear reviewer,

thanks for the time you spend to revise our work. Your suggestions are of great importance to us, in order to improve the article.

  1. We make the discussion longer adding more informations. You are right in saying that there isn't a proper review. Since it isn't our idea we deleted it from the title.
  2. We corrected this data.
  3. This is a retrospective study, without any direct intervention on the patients. We collected the data from the files of each patient and for this we obtain the consent from the Hospital and from the patients. An IRB approval isn'requested.

Round 2

Reviewer 1 Report

The authors have addressed the previous comments and overall improved the manuscript with the addition of operative details, increased data on the learning curve of the procedures, and increased discussion on the benefits of the robot with a rationale for continuing a robotic program in pediatric surgery. 

I have one other suggestion that I had not mentioned in the previous review - I assume the mortality was 0 but I would specifically say so as having a very low mortality was important for demonstrating overall safety of robotic surgery for complex operations in the adult literature.

There are some minor language/editorial edits that can be made in the new version:

1. First paragraph under the Surgery heading states "beginning of n our experience with the Da Vinci system"

2.  Second paragraph under Surgery heading states "In addition, a 5-mm laparoscopic trova using the surgeon"

3. Fourth paragraph in Discussion states "To date, the learning curve for the technique robotics in surgery"

4. 7th paragraph of discussion: "Infact robot remains an excellent aid"

5. "Robotic surgery, compared with open surgery and laparoscopy, is also shown to be better for the surgeon. In fact, especially in more complex procedures, it increases the physical fatigue of the operator, which in longer surgeries more result in an increased risk of making mistakes". I reading this paragraph with the word "it" referring to robotic surgery but I think the authors mean that open or laparoscopic surgery increases fatigue... Should clarify.

Author Response

Thanks again for the effort you put in this revision! 

We made the corrections by your indications.